# Effect of COVID-19 Lockdown on Cardiovascular Health in University Students

**DOI:** 10.3390/ijerph192315483

**Published:** 2022-11-22

**Authors:** Grzegorz Bielec, Anna Kwaśna

**Affiliations:** 1Department of Physical Culture, Gdansk University of Physical Education and Sport, ul. Gorskiego 1, 80-336 Gdansk, Poland; 2Department of Physical Education and Sport, Wroclaw University of Health and Sport Sciences in Wrocław, al. Paderewskiego 35, 51-612 Wroclaw, Poland

**Keywords:** maximal oxygen consumption, blood pressure, heart rate, students

## Abstract

Background: A decrease in physical activity levels among university students during the COVID-19 pandemic is well-documented in the literature. However, the effect of lockdown restrictions on cardiovascular fitness has not been thoroughly investigated. Methods: The aim of the study was to assess the possible changes in cardiovascular fitness among university students during a 14-week period of the COVID-19 pandemic. Thirteen female and seven male tourism and recreation students participated in the study. Examinations were conducted in November 2020 and in February/March 2021. Students performed the PWC170 test on a cycling ergometer. Maximal oxygen consumption was calculated based on the PWC170 test results. Blood pressure and heart rate were measured at rest, as well as in the 1st and 5th minute of post-exercise recovery. Results: No substantial changes were observed in maximal oxygen consumption level when comparing autumn and winter indices. Male students presented elevated blood pressure whereas female students presented normal blood pressure. Heart-rate and blood-pressure indices did not show substantial alternations in examined students during analyzed period. Conclusions: Fourteen weeks of lockdown had little effect on the cardiovascular health of tourism and recreation students.

## 1. Introduction

An attempt to stop the spread of the coronavirus disease (COVID-19) was made by tightening the national quarantine rules. The magnitude of the lockdown varied, and it affected work, education, recreation and lifestyle. Various forms of social distancing measures were introduced such as closing gyms, movie theatres, shops and fitness clubs by many countries. The lockdown, and the COVID-19 restrictions have placed limitations on opportunities for people to be physically active. Consequently, a serious problem arose regarding the potential harmful effects of inactivity. Many authors emphasized the importance of exercise for maintaining physical and mental health during the lockdown. They also mentioned that it was important in mitigating the negative health outcomes of COVID-19 [1,2]. Recent evidence also attested to the benefits of regular physical activity on survival rates for COVID-19 patients [3,4,5]. Physical activity is a well-known therapeutic tool in various diseases [6]. Physical activity can reduce the amount of cardiovascular disease and reduce the symptoms of osteoarthritis. This is why exercise can be considered a therapeutic tool in medicine [7]. Regular exercise has also been shown to be protective and beneficial in coronary heart disease, stroke, type-2 diabetes and obesity [8]. Sedentary lifestyles have been negatively correlated with cardiovascular mortality independent of age, sex and presence or lack of pre-existing cardiovascular disease [9,10].

On the other hand, limiting physical activity may contribute to the development of cardiovascular disease [11]. A meta-analysis of prospective studies, totaling 36 investigations and over three million subjects, concluded that individuals that followed the WHO guidelines for physical activity were associated with a 17% lower risk of cardiovascular events and a 23% lower risk of cardiovascular mortality [12]. According to the 2020 guidelines of the World Health Organization, for significant health benefits, adults should engage in at least 150–300 min of moderate-intensity aerobic exercise per week, or at least 75–150 min of high-intensity aerobic physical activity. Further benefits were seen in those that combined\moderate-intensity exercise and high-intensity exercise [13]. Cardiorespiratory fitness, usually expressed and measured as maximal oxygen uptake (VO_2_max), is reported in some studies as an indication of physical fitness [14,15]. When measuring this parameter, the most commonly used tests are walk/run tests, followed by cycling and step tests [16]. The maximum oxygen uptake (VO_2_max), an internationally accepted parameter to evaluate the cardio respiratory fitness reflects the amount of oxygen utilized by working muscles during maximal exercise. VO_2_max is presented in liters/min as absolute value or in milliliters/kg/min as relative VO_2_max. The results of studies conducted on people ages 18–25 during the pre-pandemic period, showed that the mean value of VO_2_max for men was 45.66 ± 8.96 mL/kg/min, and, for women, 37.85 ± 4.3 mL/kg/min [17]. Considering individual differences in VO_2_max in response to physical activity, this association should be further explored in different populations.

High blood pressure is one of the main causes of heart and vascular disorders. Whelton et al. [18] and Howden [19] et al. suggested aerobic exercise to reduce blood pressure in both hypertensive and normotensive persons. In a study conducted on 1675 university students, a direct relationship between elevated systolic blood pressure and body mass index (BMI) was found [20]. The higher the BMI, the greater the incidence of high systolic blood pressure was found. The percentage of students with a systolic blood pressure above 130 mmHg in the BMI > 25 kg/m^2^ group was 11.8%, and among those with a BMI of less than or equal to 25, only 2% had a systolic blood pressure above 130 mmHg. Similarly, in other studies, pharmacy students with the BMI value above 25 kg/m^2^ had significantly higher blood pressure values when compared to their counterparts with the BMI below 25 kg/m^2^ [21].

The coronavirus pandemic has affected many, if not all, aspects of life, including physical activity levels. Some people, when the places where they could exercise were closed, started working independently at home. More time was delegated to household chores and a sedentary lifestyle. It was found that more time was being spent in front of the TV and on the Internet than moving around and exercising. Pandemic restrictions have also changed university students’ lifestyle. Most students, after the universities were closed and e-learning was established, returned to their family homes. They were deprived of the possibility to use the recreational facilities, as well as pubs and restaurants. This situation caused a decrease in physical activity levels and decrease in satisfaction with life among university students [22]. Consequently, physical inactivity, sedentary lifestyle, and psychological factors affected their cardiovascular health [23]. When monitoring students ages 17–19, it was found that the reduction in physical activity caused by the COVID-19 restrictions led to an increase in resting heart rate, an increase in systolic blood pressure and a deterioration of the respiratory system. These changes indicate a decrease in the cardiovascular health of students [24]. In previous studies, it was shown that the decline in physical activity during the pandemic worsened the physical health and capabilities of these women. The data showed a statistically significant increase in resting heart rate, systolic blood pressure, and respiratory rate [25]. All of these markers, again, are an indication of a decline in cardiovascular and metabolic health.

In Poland, the possibilities of practicing physical activity in the open air (skate rinks, ski slopes) were significantly impeded during the so-called second wave of the pandemic. People, out of concern for their health and that of others (the sick), and fears of being infected, stayed at home. The slogan “stay at home” was widely promoted in social media and on television. This did not have a positive effect on the amount of physical activity undertaken. For this reason, in our work, we wanted to investigate whether the lockdown restrictions influenced the cardiovascular health of Polish university students. We also wanted to compare the parameters of students’ cardiovascular health taken during lockdown with literature-based findings from the pre-pandemic era. Thus, the objective of the study was to assess the changes in blood pressure, heart rate and VO_2_max in Tourism and recreation students during 14 weeks of so-called second wave of the COVID-19 pandemic. We hypothesized that (i) lockdown restrictions would negatively impact cardiovascular health in our students, and (ii) cardiovascular parameters of our students would differ from parameters of healthy university students examined in the pre-pandemic period.

## 2. Material and Methods

### 2.1. Study Design and Participants

The inclusion criteria for participation in the study were as follows: (i) self-reported good state of health, (ii) no prior SARS-CoV-2 virus infection, and (iii) the ability to participate in laboratory examinations in the local university. The study began in November 2020, just two weeks after the governmental announcement of the second COVID-19 lockdown. Remote learning was introduced to all types of schools and universities. An invitation to participate in the study was sent via Microsoft Office Teams to 67 Tourism and recreation students at the local university. Students were informed about the purpose and the course of the examination. Students were also asked to declare if they had ever been infected with the SARS-CoV-2 virus.

Initially, 31 students declared their willingness to participate in the study. Three of them had to be excluded because of previous infection with COVID-19. Eight students decided to resign from the study for health or personal reasons. Finally, twenty students (13 female and 7 male) ages 21–24 years took part in the study. Anthropometric data of examined students is presented in the “Results” section.

Examination procedures consisted of: (i) anthropomorphic measurements, and (ii) Physical Work Capacity 170 test (PWC 170). Heart rate and blood pressure were assessed before and after the PWC170 test. The first examination was conducted in November 2020, and the second one in February/March 2021, i.e., after a 14-week period. Students were asked not to change their eating habits during the study, and not to consume alcohol or caffeine for a minimum of 24 h prior to examination.

The study was conducted according to the Declaration of Helsinki. The Ethics Committee of the local university approved the study (7/2021).

### 2.2. Procedures

The measurements were conducted in the morning. Before the examination, the participants were asked about their current well-being, and their body temperature was measured with a non-contact device (Omron Gentle 720, Omron Healthcare Co., Ltd., Kyoto, Japan). All the participants declared good well-being, and their body temperature ranged from 36.4 °C to 36.8 °C. Body height was estimated with a Seca 216 stadiometer (Seca GmbH, Hamburg, Germany). Body mass, body mass index, and body fat percentage were estimated with a Tanita BC 418 MA analyzer (Tanita Corp., Tokyo, Japan). Participants were measured barefoot, and they were wearing light sports clothes. Each measurement was taken twice. Coefficient of variation was calculated for each pair of measurements, and its value varied from 2.1 to 3.1.

After the anthropomorphic measurements were completed, the participants had their heart rate, and blood pressure measured in a sitting position. The Omron M3 Comfort automatic blood pressure monitor (Omron Healthcare Co., Ltd., Kyoto, Japan) was used as a tool.

Next, the participants underwent the Physical Work Capacity 170 test (PWC 170) on a Monark 874-E cycloergometer (Monark, Vansbro, Sweden). The test started with a one-minute cycling warm-up with a load of 30 W. Participants kept on pedaling constantly for 5 min with a load of 1 W per 1 kg of body mass. After a one-minute rest period, the participants continued pedaling for another 5 min with a load of 1.5 W per 1 kg of body mass. Participants were instructed to pedal with a steady cadence of 50 revolutions per minute. The test finished after a one-minute cool-down with a load of 30 W. Participants wore a Polar H10 heart rate monitor (Polar Electro, Kempele, Finland) during the testing procedure. The maximum heart rate value was recorded at the 12th minute of the test. After completing the PWC 170 test, the heart rate, and blood pressure were measured during the 1st, and the 5th minute of post-exercise recovery period. All the measurements were conducted by an experienced researcher.

### 2.3. Calculations

Maximal oxygen consumption (VO_2_max) was calculated indirectly based on the PWC 170 test results. The following formula was utilized:VO_2_max = 1.7 × PWC170 + 1240
where 1.7 and 1240 are still seizes, and PWC170 is a test result expressed in kGm/min [26].

Statistical calculations were performed with Statistica 12.0 software (StatSoft, Tulsa, OK, USA). The results of the Shapiro–Wilk test confirmed the normality of the analyzed data. Analysis of variance (ANOVA) and the post-hoc Tukey test were used to detect the differences in heart rate and blood pressure variations in both terms of the examination. A paired *t*-test was used to calculate the differences between the VO_2_max values measured in November and in February/March.

## 3. Results

Table 1 presents the anthropomorphic data of examined students.

Neither male nor female students demonstrated substantial changes in their body composition in the analyzed period.

Table 2 presents the changes in maximal oxygen consumption in examined students. No significant changes in VO_2_max occurred in female and male students during 14 weeks of lockdown.

Figure 1 reflects the heart-rate variations in female students during testing procedures in November and in February/March. Heart-rate variations had very similar patterns in both dates of examination. Female students reached the highest HR values in the 10th minute of the PWC170 test. Heart-rate returned to pre-test values during the 5th minute of post-exercise recovery. The post-hoc Tukey test did not reveal significant changes between heart-rate values measured in November and in February/March.

Figure 2 presents heart-rate alternations in male students. Similarly to their female counterparts, male students demonstrated the highest HR values in the final phase of the PWC170 test.

Heart-rate values at the 5th minute of post-exercise recovery were even lower than in pre-test conditions, but these changes were insignificant. According to the Tukey post-hoc test, heart-rate variations did not differ substantially between November and February/March measurements.

Figure 3 displays the changes in systolic and diastolic blood pressure in female students during analyzed period. Systolic blood pressure value rose after the cycling test, and reduced during the recovery period. Systolic blood pressure values were insignificantly lower in the 5th minute of post-exercise recovery when compared to pre-test indications. Conversely, diastolic blood pressure values reduced immediately after the cycling test. In autumn measurements diastolic blood pressure decreased on the 5th minute of post-exercise recovery to below the pre-test values. In contrast, during winter measurements diastolic blood pressure rose in the 5th minute of recovery above the pre-test values. However, these changes were insignificant.

Figure 4 presents blood pressure variations in male students. The course of changes is different when measurements in November, and in February/March are compared. During autumn measurements there were no significant alternations in systolic blood pressure thorough the entire testing procedure. In contrast, during winter measurements systolic blood pressure rose after the cycling test, and significantly reduced in the 5th minute of post-exercise recovery. The highest value of diastolic blood pressure (84 mmHg) was noted in pre-test conditions in November, and the lowest (75 mmHg) in the 5th minute of post-exercise recovery in February/March session. Diastolic blood pressure did not show significant variations in both terms of examination.

## 4. Discussion

### 4.1. Discussion of the Results

The current study aimed to identify possible changes in cardiovascular parameters among university students during the second COVID-19 lockdown. During the first examination (November 2020), based on blood pressure results, our female students were classified as normotensive, whereas our male students presented stage 1 hypertension [27]. In the analyzed period, we observed no substantial changes in heart-rate variations, blood-pressure variations, and oxygen uptake in the examined students. However, in the second examination (February/March 2021) male students were classified as having elevated blood pressure. The results of previous studies conducted among Polish university students also show the higher values of blood pressure in male students when compared to their female counterparts [28]. The elevated blood pressure assessed during rest in our male students might have been caused by stress before exercise testing. This phenomenon is associated with incident hypertension regardless of risk factors, and occurs in normotensive men [29]. Possibly, the elevated blood pressure results in our male students could also have been affected by stress associated with laboratory measurement procedures. This sort of stress is known as “white coat syndrome” and occurs in both hypertensive and normotensive subjects [30,31]. It is plausible that our male students accustomed themselves to laboratory conditions during the second examination in February/March; therefore, their resting blood pressure was lower when compared to the November session.

To the best of our knowledge, the current study is the first attempt to estimate cardiovascular fitness among healthy university students during the pandemic lockdown. The literature presents some findings concerning the cardiovascular health of older adults in the pandemic era. For example, a study conducted among middle-aged healthy adults in the first lockdown (spring 2020) revealed a significant increase in blood pressure as a result of an unhealthy lifestyle [32]. Conversely, the results of another study showed substantial decrease in blood pressure among middle-aged men and woman during the first lockdown period [33]. One observational study displayed non-significant changes in VO_2_max among army recruits who were not diagnosed with SARS-CoV-2 infection. On the other hand, recruits who suffered from COVID-19, showed significant declines in their VO_2_max level [34].

Numerous studies confirmed the decrease in physical activity levels among university students during the COVID-19 outbreak [35,36]. However, our previous observations with the same group of tourism and recreation students showed no significant changes in their physical activity levels in a 14-week period of COVID-19-related lockdown [37]. According to the International Physical Activity Questionnaire results our students’ responses indicated moderate levels of physical activity during the lockdown period. In the current study, our students (both female and male) demonstrated similar VO_2_max values to those of physical education (PE) students examined in the pre-pandemic period [38,39]. Although no substantial changes in VO_2_max levels occurred among our students in the analyzed period, our female students presented higher VO_2_max values than their police academy counterparts when assessed before the pandemic era [40]. Discrepancies in the abovementioned results might have been affected by different testing protocols (step test vs. cycling test). We suppose that the lack of changes in VO_2_max levels in our students is an effect of insufficient physical activity being undertaken during the lockdown period. Similar conclusions came from the study conducted by Oue et al. [41] in the pre-pandemic era. In their study, no significant changes were observed in physical work capacity level in inactive university students before and after an 8-week period. On the other hand, a significant increase was observed in physical work capacity among students who underwent continuous or interval exercise training for 8 weeks. It is worth mentioning that, in the current study, both female and male students presented good cardiovascular fitness according to McArdle’s et al. classification [42].

To the best of our knowledge, no data concerning the cardiovascular response to physical effort in university students during lockdown are presented in the literature. Thus, the comparisons of our results refer to studies conducted before the pandemic era. For example, in both dates of our examinations, our male students presented higher blood-pressure values in resting conditions when compared to male physical education students (133/84 mmHg and 129/78 mmHg vs. 124/81 mmHg) [43]. Similarly, the resting blood pressure levels of our female students differed when compared to female PE students in another study (119/75 mmHg in autumn and 117/74 mmHg in winter vs. 110/72 mmHg) [44]. This difference can be explained by the possibly higher level of cardiorespiratory fitness shown in PE students. In another study, PE female students had their blood pressure and heart-rate measured before, during, and five minutes after a 10-min running test. In these students, their blood pressure rose from 111/75 mmHg (pre-) to 114/77 mmHg (post-) [45]. In contrast, the blood pressure of our students decreased from 119/75 mmHg to 112/71 mmHg and from 117/74 mmHg to 113/75 mmHg during autumn and winter examinations, respectively. Our male students also presented lower blood pressure values at the 5th minute of post-exercise recovery when compared to the pre-exercise conditions (133/84 mmHg vs. 128/77 mmHg in autumn, and 129/78 mmHg vs. 120/75 mmHg in winter). The decline in blood pressure after physical exertion is a typical reaction of the cardiovascular system to the effort-induced stress. Decreased blood pressure values may last for up to 24 h when compared to the pre-exertion period [46]. Cardoso et al. [47] reported hypotension effect as a result of single aerobic exercise; however, this effect was more distinct in subjects presenting elevated blood pressure. We observed a similar situation in our study, where male students significantly decreased their systolic blood pressure after the PWC 170 test during the winter examination. The differences between blood pressure indices presented in the current paper and those presented in the abovementioned studies cannot be explained unequivocally, as many factors influence cardiovascular response for physical effort. The literature review indicates a relationship between exercise characteristics (i.e., duration, intensity, repeatability) and post-exercise hypotension effects [48,49]. We suppose that the different blood pressure results described in the discussed papers were affected by the various exercise modes used in physical examination.

Regarding different testing protocols, the literature provides various types of data concerning the heart-rate response for maximal and submaximal effort in university students. Our male students demonstrated distinctly higher HR values in resting conditions (pre-test) compared to trained and non-trained students of sport sciences [50]. For maximal heart-rate values, in a multi-staged cycling test, non-athlete students had an HRmax of 158 bpm (males) and 170 bpm (females) [51]. On the other hand, male athlete students achieved an HRmax value of 188 bpm in a maximal incremental test on a kayak ergometer [52]. In contrast, our male and female students had an HRmax of 140 bpm and 153 bpm, respectively. The results of another study showed that female physical education students had an HRmax of 112 bpm, while female medical students had an HRmax of 126 bpm, as a result of 10-min submaximal running test [45]. Heart-rate recovery also shows alternations regarding the various exercise tests protocols. Berry et al. [53] reported that male and female students who underwent an incremental test to exhaustion had heart-rates of 162 bpm and 160 bpm in the first minute of post-exercise recovery. In the fifth minute of post-exercise recovery both male and female students had HR values of 105 bpm. In our study, during both examination dates, male students had HR values of 106 bpm, and HR values of 96 bpm in the first and fifth minute of post-exercise recovery, respectively. As a comparison, our female students had HR values of 105 bpm (autumn) and 108 bpm (winter) in the first minute of post-exercise recovery, and 100 bpm (autumn) and 84 bpm (winter) in the fifth minute of post-exercise recovery. The observed discrepancy in the compared results is probably caused by the different protocols used in the exercise tests. On the other hand, athlete and non-athlete students who underwent the same graded treadmill test, did not differ significantly in terms of heart rate recovery in the first minute after physical effort [54].

### 4.2. Strengths and Limitations

The small number of examined students is a limitation of the current study. Thus, the presented results should be interpreted with caution. We were not able to gather a large number of participants due to the lockdown conditions. During the second wave of the COVID-19 pandemic, the students’ dormitories were closed and all classes were exclusively conducted online. On the other hand, the measurements taken in direct contact with the small number of participants gave us reliable outcomes. We believe that having these direct contacts is a strength of our study. Another strength of our study is a reliable insight into the cardiovascular health of university students during the lockdown period. This is probably the only examination of cardiovascular parameters in university students conducted in laboratory conditions. We believe that our study provides beneficial data to the other researchers in this field. Nevertheless, our results should not be generalized because of obvious differences between lifestyle conditions before and during the COVID-19 pandemic.

## 5. Conclusions

Our study showed that 14-week lockdown restrictions concerning remote learning, and sports and recreation facilities’ attendance, did not affect the cardiovascular health of tourism and recreation students. The maximal oxygen uptake level did not show substantial alternations during the analyzed period. Tourism and recreation students demonstrated good cardiovascular fitness and their VO_2_max results were comparable to the results demonstrated by healthy university students before the pandemic era. Exercise-induced heart rate and blood pressure indices showed some diversity when compared to the results of healthy students examined before the lockdown period. These differences were probably due to the different physical examination protocols used in each study.

## Figures and Tables

**Figure 1 ijerph-19-15483-f001:**
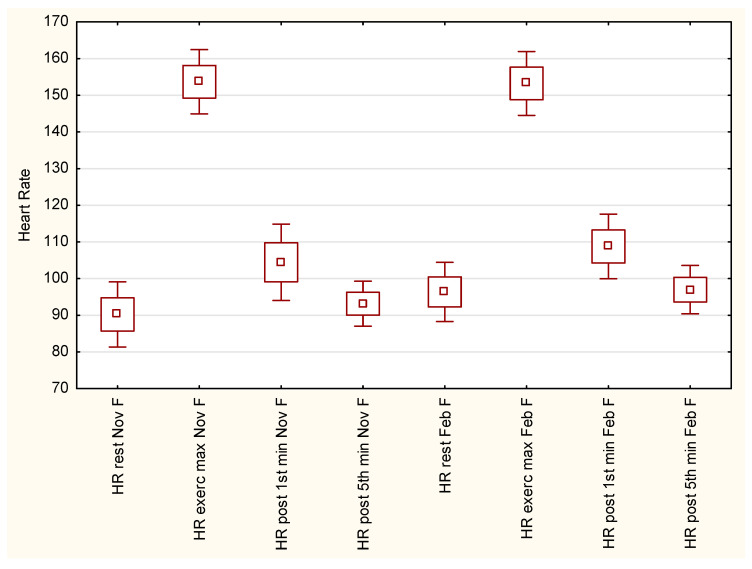
Heart-rate variation in female students in autumn and winter sessions of Physical Work Capacity 170 test.

**Figure 2 ijerph-19-15483-f002:**
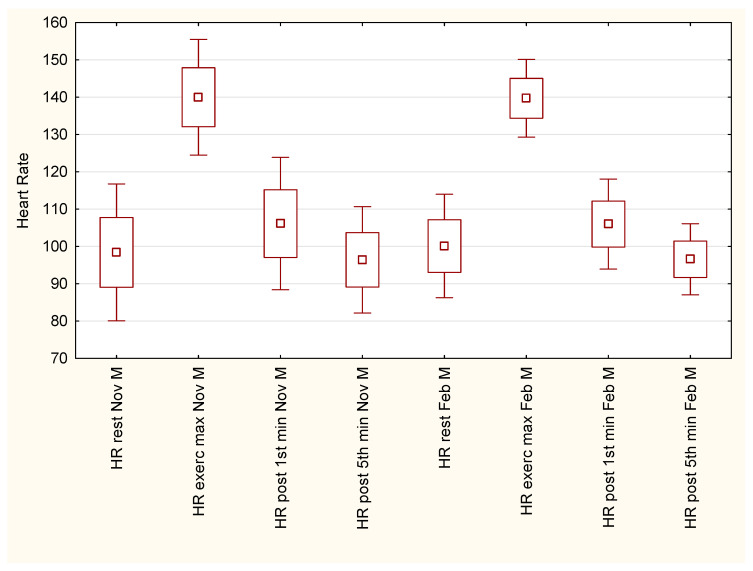
Heart-rate variation in male students in autumn and winter sessions of Physical Work Capacity 170 test.

**Figure 3 ijerph-19-15483-f003:**
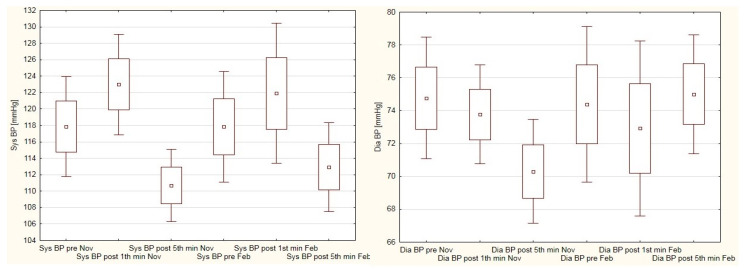
Blood pressure variations in female students in autumn and winter sessions of Physical Work Capacity 170 test.

**Figure 4 ijerph-19-15483-f004:**
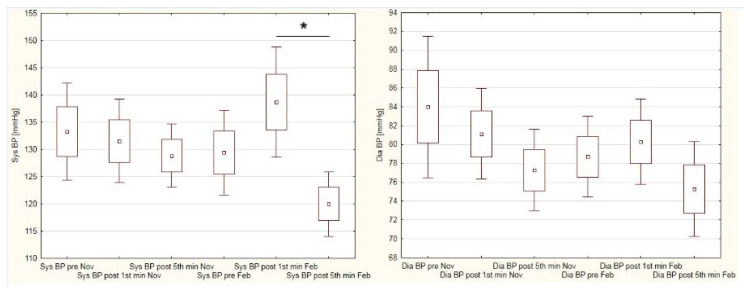
Blood pressure variations in male students in autumn and winter sessions of Physical Work Capacity 170 test. * *p* < 0.05.

**Table 1 ijerph-19-15483-t001:** Body composition in examined students in 14-week period of COVID-19 pandemic. Data is presented as mean value ± standard deviation.

	Body Height [cm]	Body Mass [kg]		Body Mass Index [kg/m^2^]		Body Fat [%]	
		Pre	Post	Pre	Post	Pre	Post
Females (*n* = 13)	166.3 ±5.5	58.5 ±9.7	58.6 ±10.2	21.0 ±2.6	21.0 ±2.7	24.1 ±6.6	24.1 ±7.3
*t*-test *p* value		0.85		0.86		0.87	
Males (*n* = 7)	176.7 ±7.9	69.0 ±8.6	69.9 ±9.4	22.1 ±2.3	22.3 ±2.5	17.2 ±5.6	16.8 ±5.7
*t*-test *p* value		0.06		0.18		0.80	

**Table 2 ijerph-19-15483-t002:** Maximal oxygen consumption in examined students in 14-week period of COVID-19 pandemic. Data are presented as mean value ± standard deviation.

	VO_2_max [mL/min]		VO_2_max [mL/kg/min]	
	Pre	Post	Pre	Post
Females (*n* = 13)	2402.39 ± 272.37	2467.72 ± 72	41.79 ± 6.4	43.28 ± 11.11
*t*-test *p* value	0.254		0.215	
Males (*n* = 7)	3172.6 ± 725.53	3184.2 ± 553.15	46.69 ± 12.03	46.07 ± 8.41
*t*-test *p* value	0.475		0.418	

## Data Availability

The data presented in this study are available on request from the corresponding author.

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
