# Peer review of "Effect of COVID-19 Lockdown on Cardiovascular Health in University Students"

_ijerph, 2022, doi:10.3390/ijerph192315483_

Round 1

Reviewer 1 Report

Thank you for the opportunity to review this article. Unfortunately the COVID-19 pandemic is not over, so the topic is topical.

Introduction

More information on the pandemic and the lockdown.

The research hypothesis and objectives are present and correctly described.

Material and Methods

Study design and participants

It is not clear to me whether there was a way to control any asymptomatic infection among participants. Were blood tests performed to determine the presence of antibodies?

Results

The results are presented logically and coherently.

Discussion

The discussions are well outlined according to the results presented.

Conclusions

I think more information is needed in the conclusions section.

I am of the opinion that the limitations should be extracted from the discussion section and placed in a separate chapter.

Also, as far as I can see in the study, psychological factors were not taken into account as a potential vector in terms of determining the increase in blood pressure. Numerous studies have shown that psychological stress can lead to increased heart rate and blood pressure. However, the onset of the pandemic has been blamed for an alarming increase in anxiety and depression.

Were students assessed at the beginning and end of the study for stress/anxiety/depression?

The sample size is small, but the authors are aware of this. However, some paragraphs present the findings as if they were general, which is not the case.

Author Response

On behalf of authors of the article, I would like to thank you for objective and thorough review. Proper comments were taken into account and we believe there were contribute to improve the scientific level of the paper. After receiving reviews our intention was to incorporate changes that were marked with red color in the text.

Thank you for the opportunity to review this article. Unfortunately the COVID-19 pandemic is not over, so the topic is topical.

Introduction

More information on the pandemic and the lockdown.

More information about the effect of lockdown on university students' lifestyle were added to the "Introduction" section

The research hypothesis and objectives are present and correctly described.

Material and Methods

Study design and participants

It is not clear to me whether there was a way to control any asymptomatic infection among participants. Were blood tests performed to determine the presence of antibodies?

Before the examination we asked each participant about his/her current well-being. Additionally we measured the body temperature of each participant. These information were added to the "Material and Methods" section. In fact, we had neither the equipment, nor the competencies to perform blood tests.

Results

The results are presented logically and coherently.

Discussion

The discussions are well outlined according to the results presented.

Conclusions

I think more information is needed in the conclusions section.

We put more information summing up our results in the "Conclusions" section.

I am of the opinion that the limitations should be extracted from the discussion section and placed in a separate chapter.

The "Strength and limitations" sub-section was created according to your suggestion.

Also, as far as I can see in the study, psychological factors were not taken into account as a potential vector in terms of determining the increase in blood pressure. Numerous studies have shown that psychological stress can lead to increased heart rate and blood pressure. However, the onset of the pandemic has been blamed for an alarming increase in anxiety and depression.

We followed your suggestion and discussed the probable factors elevating blood pressure in laboratory conditions. These information were added to the "Discussion" section.

Were students assessed at the beginning and end of the study for stress/anxiety/depression?

We did not conducted any stress/anxiety/depression tests among students. As we wrote above, we asked the participants about their current well-being before examination.

The sample size is small, but the authors are aware of this. However, some paragraphs present the findings as if they were general, which is not the case.

In the "Strength and limitations" sub-section we stated that our results should not be generalized because of completely different circumstances accompanying lockdown period.

Reviewer 2 Report

Lines 110 to 111; "consisted of..." I didn't understand the difference (ii. PWC170 test, vs iii. PWC170 test) between point (ii) and point (iii). I suggest being more specific.

I suggest including in table 1 data like type of exercise, time of exercise done

Lines 257 to 271; "Similarly..." These paragraphs are interesting, but it seems the authors only describe results between studies and their results, I recommend do hypotheses between the results of your students and other publication, and also describing why the levels of pressure level were minor in autumn and winter.

It is possible to include if the exercise was an important variable related to the results

Overall comments

This study is interesting and well done. The authors made a great effort to do it. I have some suggestions that the authors need to resolve.

Author Response

On behalf of authors of the article, I would like to thank you for objective and thorough review. Proper comments were taken into account and we believe there were contribute to improve the scientific level of the paper. After receiving reviews our intention was to incorporate changes that were marked with red color in the text.

Lines 110 to 111; "consisted of..." I didn't understand the difference (ii. PWC170 test, vs iii. PWC170 test) between point (ii) and point (iii). I suggest being more specific.

We did some corrections in these lines, and we believe that present version is more clear for readers.

I suggest including in table 1 data like type of exercise, time of exercise done

Unfortunately we did not control specific forms of physical activity undertaken by our students. Physical activity levels of the students were assessed according to the results of the International Physical Activity Questionnaire (IPAQ). No specificity of physical exercises is included in the IPAQ questions. We attach our previous article, where the IPAQ results are presented.

Lines 257 to 271; "Similarly..." These paragraphs are interesting, but it seems the authors only describe results between studies and their results, I recommend do hypotheses between the results of your students and other publication, and also describing why the levels of pressure level were minor in autumn and winter.

It is possible to include if the exercise was an important variable related to the results

According to your suggestion we created hypotheses in the end of the "Introduction" chapter. We referred to these hypotheses in the "Discussion" section, and we cited more appropriate literature. We also discussed the theme of exercise as a possible variable related to the results.

Overall comments

This study is interesting and well done. The authors made a great effort to do it. I have some suggestions that the authors need to resolve.

Round 2

Reviewer 1 Report

Thank you for the opportunity to review this article.

The authors have taken my recommendations into consideration. I agree with this revised form of the article.